# Molecular Epidemiology of Azole-Resistant *Aspergillus fumigatus* in Sawmills of Eastern France by Microsatellite Genotyping

**DOI:** 10.3390/jof6030120

**Published:** 2020-07-26

**Authors:** Audrey Jeanvoine, Chloé Godeau, Audrey Laboissière, Gabriel Reboux, Laurence Millon, Steffi Rocchi

**Affiliations:** 1Parasitology-Mycology Department, University Hospital of Besançon, 25000 Besançon, France; ajeanvoine@chu-besancon.fr (A.J.); gabriel.reboux@univ-fcomte.fr (G.R.); lmillon@chu-besancon.fr (L.M.); 2Chrono-Environnement UMR 6249 CNRS, Bourgogne Franche-Comté University, 25000 Besançon, France; chloe.godeau@univ-fcomte.fr (C.G.); audrey.laboissiere@univ-fcomte.fr (A.L.)

**Keywords:** *Aspergillus fumigatus*, azole-resistant, sawmill, microsatellite, multilocus genotype

## Abstract

**Background:** Wood chipping has been described as a potential hotspot for the selection of azole-resistant *Aspergillus fumigatus* (AR*Af*). We previously reported AR*Af* isolates in sawmills (Eastern France), most of which contained the TR_34_/L98H mutation. **Methods:** To study genotypic relatedness, microsatellite genotyping (short tandem repeat for *A. fumigatus* (STR*Af*)) was performed on 41 azole-susceptible *A. fumigatus* (AS*Af*) and 23 AR*Af* isolated from 18 sawmills and two clinical *A. fumigatus* (sensitive and resistant) isolated from a sinus sample of a woodworker. **Results:** Fifty-four unique multilocus genotypes (MLGs) were described among the 66 isolates: 13/24 AR*Af* and 41/42 AS*Af.* Allelic diversity was higher for AS*Af* than for AR*Af*. Among the 24 AR*Af*, five isolates had their own MLGs. Thirteen AR*Af* (54%) belonged to the same group, composed of four close MLGs, defined using Bruvo’s distance. Thirty-two of the 42 AS*Af* (76%) had their own MLGs and could not be grouped with the Bruvo’s distance cutoff used (0.2). **Conclusion:** Thus, at a regional scale and in the particular environment of the wood industry, common but also different distinct genotypes, even in the same sawmill, were identified. This suggests that the hypothesis of AR*Af* clonal expansion from a common strain is probably insufficient to explain genotype emergence and distribution.

## 1. Introduction

*Aspergillus fumigatus* is a saprophytic fungus that is widespread in the environment with an ecological niche of decaying vegetation and soil [1]. It is also a ubiquitously opportunistic pathogen responsible for aspergillosis, notably invasive aspergillosis (IA)—the most severe form of the disease [2]. Indeed, the IA mortality rate is high and can reach 65% [3].

Although azole antifungals improve the management of *Aspergillus*-disease, this clinical advance might be threatened by the emergence of azole-resistant *A. fumigatus* (AR*Af*) worldwide [4,5]. Over the past 20 years, there have been increasing reports of AR*Af* recovered on the five continents both in clinical and environmental samples [4]. The environmental route of resistance, based on the wide scale use of azole fungicides in the environment, plays a significant role [6]. Two main mechanisms of resistance that are probably of environmental origin, TR_34_/L98H and TR_46_/Y121F/T289A mutations, have commonly been described in strains from azole-naïve patients and from the environment [4,5].

Over the past few years, several environmental areas have been reported as potential hotspots for the selection of AR*Af*. In the Netherlands, composts containing azole residues constitute a potential hotspot for the emergence of new mutations conferring resistance [7]. Recently, wastes originating from flower bulbs, green materials, and wood chippings have been reported to be three new hotspots containing azole fungicides with the highest proportion of AR*Af,* as compared to wheat cereals, animal manure, grain, maize silage, fruit storage or regional and exotic fruit waste [8]. AR*Af* have been identified in wood environments for some years now. In fact, AR*Af* carrying TR_34_/L98H and TR_46_/Y121F/T289A have been described in wood debris of tree trunk hollows in Tanzania and Romania [9]. We also previously described the presence of AR*Af* in the wood environment, namely sawmills of Eastern France, most of which carried the TR_34_/L98H mutation [10]. In this previous study, the presence of AR*Af* carrying the TR_34_/L98H mutation seemed to greatly depend on the azole fungicide formulation and the quantity in substrates [10].

Consequently, the azole fungicide selection pressure could occur in different environments and in different countries and seems to play an important role in the selection of AR*Af*. Although typing of *A. fumigatus* isolates could provide insights into the dynamics of azole resistance development, the origin and global diffusion of azole-resistant *A. fumigatus* with TR_34_/L98H and TR_46_/Y121F/T289A are still unclear [11,12]. Currently, short tandem repeat for *A. fumigatus* assay (STR*Af*), based on a nine microsatellite analysis, is accepted as the reference and robust typing method for this species [13,14].

The aim of this study was to genotype AR*Af* and azole-susceptible *A. fumigatus* (AS*Af*), previously isolated from sawmills of Eastern France, by using STR*Af* typing in order to study their genetic relatedness and to describe the potential clustering of isolates according to their susceptibility profile [10]. In addition, the clinical isolates (sensitive and TR_34_/L98H) previously described in an immunocompetent woodworker with invasive sinusal aspergillosis were compared to the sawmill isolates [15].

## 2. Materials and Methods

### 2.1. Collection of A. fumigatus Strains

A total of 64 environmental *A. fumigatus* strains, isolated from 18 French sawmills (Eastern France) between September 2014 and April 2016, were analyzed (Table 1). Among them, 23 AR*Af* isolates had previously been isolated by using two selective homemade malt extract agar (ThermoFisher, Waltham, MA, USA) media containing azole antifungals: itraconazole called “Maltitra” and voriconazole called “Maltvori” [10]. The 41 selected AS*Af* isolates matched according to the location of AR*Af* isolates for each sawmill. Two clinical isolates (one AR*Af* and one AS*Af*), isolated from sinus samples in an immunocompetent woodworker who developed an invasive sinusal aspergillosis following facial injuries from a work-related accident, were also included (Table 1) [15].

All *A. fumigatus* isolates were identified to the species level by PCR amplification and sequencing a part of the highly conserved β-tubulin gene [10].

For all AR*Af*, minimal inhibitory concentrations (MIC) of four medical azoles (itraconazole, voriconazole, posaconazole and isavuconazole) and two azoles used in agriculture and sawmills (propiconazole and tebuconazole) were obtained by using the EUCAST (European Committee on Antimicrobial Susceptibility) microdilution method [16].

Amplification and sequencing of the *cyp51A* gene were performed as previously described [10]. Among the 23 environmental AR*Af,* 20 carried the TR_34_/L98H mutation, two the TR_34_/L98H/S297T/F495I mutation and one the P216L mutation [10]. The clinical AR*Af*, one of the two isolates recovered in the woodworker, also carried the TR_34_/L98H mutation [15].

MIC of voriconazole and itraconazole for environmental and clinical AS*Af* isolates were measured by the EUCAST method [16].

### 2.2. Short Tandem Repeat for A. fumigatus (STRAf) Typing and Analysis

STR*Af* typing is based on the amplification of nine highly polymorphic microsatellite markers (STR*Afs* 2A, 2B, 2C, 3A, 3B, 3C, 4A, 4B, 4C). These nine STR*Af* loci were amplified with three triplexed PCRs as previously described [13].

PCR products were diluted 10-fold with water for molecular biology and then analyzed on the Applied Biosystems 3130 Genetic Analyzer (ThermoFisher, Waltham, MA, USA) with GeneScan™ 400 HD ROX™ size standard according to the manufacturer’s instructions (Thermofisher^®^, Waltham, MA, USA).

Amplicon sizes were determined with GeneMapper software (version 5, ThermoFisher, Waltham, MA, USA) and then transformed in repeat number.

STR*Af* analysis was then performed using RStudio software (version 3.2.2, Boston, MA, USA). Microsatellite genotype distances were calculated using Bruvo’s distance (cutoff value = 0.2) and a minimum spanning network was calculated via the bruvo.msn function on the poppr library [17]. Finally, allelic diversity was calculated for the nine STR*Af* loci for AR*Af* and AS*Af* isolates by using the Simpson index of diversity (*D*).

## 3. Results

A total of 54 STR*Af* unique multilocus genotypes (MLGs) were described among the 66 typed *A. fumigatus*: 13 for the 24 AR*Af* and 41 for the 42 AS*Af.* None of the AR*Af* isolates had MLGs close to those that of the AS*Af* isolates, and the sensitive clinical isolate identified in the woodworker before the resistant one revealed a completely different MLG.

The nine microsatellite markers including STR*Af* genotypes, allele count, repeat range, median repeat number and allelic diversity are summarized in Table 2. All loci were polymorphic with a number of alleles ranging from 3 (loci 3B and 4B) to 11 (locus 3A) and from 9 (locus 4B) to 19 (loci 3A and 3C) for AR*Af* and AS*Af* isolates, respectively. Loci 3A and 3C displayed the highest polymorphism and allelic diversity for both AR*Af* and AS*Af* isolates, whereas locus 4B was the least discriminating with the lowest number of alleles (Table 2). For all microsatellite markers, allelic diversity was higher for AS*Af* isolates (average *D* = 0.811) than for AR*Af* isolates (average *D* = 0.518) (Table 2).

Among the 42 AS*Af*, 41 MLGs were identified, with two isolates from two different sawmills (A and D, 60 km away) sharing the same genotype (A177 and D159, Figure 1). Thirty-two MLGs (76%) were very distinct and could not be grouped with the Bruvo distance cutoff used. However, four groups of two AS*Af* were found: D161 and R12P, E21P and O166, S179 and C1P, F20P and F19P (genotypes connected by lines in Figure 1). These AS*Af* were isolated from different sawmills, except for isolates F20P and F19P which were recovered in the same sawmill (F) at different locations.

Among the 24 AR*Af*, 13 MLGs were characterized. Four isolates with the TR_34_/L98H mutation (genotype numbers Q10, M18, D8 and F13, Figure 1) and one with the P216L mutation (genotype S24) had their own MLGs and could not be grouped with Bruvos’s distance. The others were recovered for several isolates grouped in the same circles within the minimum spanning network (Figure 1).

For TR_34_/L98H isolates, two MLGs that were genotypically distinct from others were identified: genotypes D3 and D4 with the TR_34_/L98H mutation, found in the same soil sample, and D5 and F14 with the TR_34_/L98H/S297T/F495I, taken from two geographically distant sawmills (170 km).

Moreover, two groups of MLGs were found with Bruvo’s distance (Figure 1). The first group was composed of four MLGs, including 13/24 AR*Af* (54%, genotype numbers A25, R21, G15, P12 / D2, D7, S23, O20, P11 / D6, O19, H16 and K17). Among these four closely related STR*Af* genotypes, three differed by only one repetition of the microsatellite marker 3A; the fourth (K17 AR*Af*) differed from the three others by the number of repetitions of microsatellite markers 3A and 3C. These four genotypes included TR_34_/L98H AR*Af* isolated from different sawmills that could be located as far as 200 km from each other; two of these genotypically similar AR*Af* (P11 and P12) were isolated in one sawmill that imported wood from Russia. The second group consisted of one clinical TR_34_/L98H isolate responsible for invasive sinusal aspergillosis in an immunocompetent woodworker and had a genotype close to a sawmill AR*Af* also carrying the TR_34_/L98H mutation (genotype R22 in Figure 1). It differed by the number of repetitions on microsatellite markers 3A and 4A. However, the woodworker did not work in sawmill R.

Identical MLGs were thus found in different sawmills, but different MLGs were also found in one sawmill: five different MLGs were identified among the seven AR*Af* found in sawmill D. This sawmill shared one MLG (genotype D5, mutation TR_34_/L98H/S297T/F495I) with another sawmill. Sawmill D shared MLGs with other sawmills and genotypes that closely resembled each other (genotypes D2, D7 and D6). It also had one MLG (genotypes D3 and D4) that was different from all other MLGs.

## 4. Discussion

In this study, we reported the presence of both common and different genotypes in AR*Af* and AS*Af* from sawmills of Eastern France by using STR*Af* typing. Our findings are similar to those of several other studies, as we reported that AS*Af* had a greater genotypic diversity than AR*Af* [14,18,19]. This could suggest a predominantly clonal expansion of AR*Af* in the environment. In fact, contrary to AS*Af*, more than half of AR*Af* (54%) belonged to the same group with the Bruvo’s distance cutoff used and seemed to be genotypically close. These AR*Af* came from different sawmills that were geographically far from one another (up to 200 km) and for which a selection pressure by azole fungicide was reported. Direct or indirect contact with azole fungicide was shown, and azole fungicides were detected in substrate samples in these sawmills [10].

This study has some limitations. First, we describe here the genotypic structure of isolates coming from only one region in France and one type of environment. A broader study including genotyping of isolates from other regions, environments and countries would make it possible to know if the genotypes of AR*Af* and AS*Af* from Eastern France are shared by isolates in other regions of the world; this could help to improve our understanding of the emergence, diffusion and distribution of AR*Af.*

In addition, we used STR*Af* typing which is widely accepted as the reference typing method for *A. fumigatus* [14]. However, two of the nine microsatellite markers, 3A and 3C, show a lower level of stability which has to be taken into account when interpreting STR*Af* data during molecular epidemiological analysis [20]. This is illustrated by the greater polymorphisms for these two markers described in our study. Moreover, three genotypes differed by only one repetition on marker 3A and grouped isolates coming from the same sawmill or the same samples (P11 and P12). The difference between these three genotypes might be valid but we cannot exclude a bias in the genotyping method used. Despite the high discriminatory power of STR*Af* typing, it remains lower than that of whole genome sequencing (WGS). So, it would be interesting to determine whether or not AR*Af* with the same genotype, revealed using STR*Af* typing, would reveal exactly the same genotype when using WGS. Likewise, exploring genotypic diversity with WGS could be useful.

It is important to emphasize that 13 STR*Af* genotypes were found among the 24 AR*Af*. Moreover, some AR*Af* had their own genotypes and were genotypically distinct from others despite sharing the same mechanism of resistance (mutation TR_34_/L98H). It is interesting to note that this diversity was reported in one sawmill (D), without any particular characteristics when compared to other sawmills using a dip processing tank, and where five MLGs were reported among the seven AR*Af* genotyped. These findings contrast with those of some studies describing only one clone of TR_34_/L98H AR*Af* strains [18,21]. The local diversity of the genotype reported here was also highlighted for AR*Af* isolates in market gardens of the same region in Eastern France. In this study, 22 STR*Af* genotypes among 44 analyzed AR*Af* and six groups of genotypically close AR*Af* were recorded. AR*Af* isolates could be more diverse in other countries such as Colombia where 19 MLGs were identified in 21 AR*Af* isolates in the environment [17,22]. Additionally, it has also been shown by STR*Af* typing that resistant isolates, coming from several countries, belonged to several groups. Similarly, some authors have reported a dispersed structure in AR*Af* with tandem repeat (TR) mutations [11,14]. Thus, it seems unlikely that the emergence of azole-resistant strains could be the work of a single clone.

The population structure of *A. fumigatus* is very complex and is likely to be due to several past and current events: historical differentiation, contemporary gene flow, sexual reproduction, recombination, and the localized azole fungicide selection that could drive expansion of AR*Af* genotypes [11,12]. It was recently reported that certain practices like composting (i.e., stockpiling plant waste) might be the key to resistance selection in *A. fumigatus* [8]. Some environments supporting the growth, sexual reproduction, genetic variation and containing residues of azole fungicides could cause these complex mutations to emerge, amplify and spread. So, AR*Af* may very well emerge in specific environments and spread to other countries due to natural factors such as wind, or anthropogenic factors such as human travel and commercial trade [11,23]. This gene flow could explain the fact that genotypically close AR*Af*, or those having the same genotype, were found in different countries [18,19,21].

A comparison of genotypes from sawmill AR*Af* with genotypes from other *A. fumigatus* isolates in a London database (https://afumid.shinyapps.io/afumID/, 4049 *A. fumigatus* isolates collected worldwide) showed that some sawmill AR*Af* (A25, R21, G15, P12 with the same MLG) presented the same genotype as a clinical Australian isolate (data not shown). Among them, isolate P12 is one of the two isolates found in one sawmill that did not treat wood, but rather imported foreign wood from Russia that had already been treated with fungicides (sawmill P). Curiously, these two AR*Af* shared the same MLG as other sawmills AR*Af*. According to these results, the gene flow could not be excluded here and it may be possible that these AR*Af* emerged in a favorable environment outside of France and then, when imported, spread to the Eastern France sawmills. Conversely, an MLG identified twice in a single sawmill (D) is not linked to any profile in the London database.

The distribution of sawmill AR*Af* and AS*Af* genotypes, with a lower diversity of AR*Af* isolates, also seems to concur with two hypotheses reported by other authors to explain the emergence and distribution of AR*Af*: the specific genetic background and predisposition of some clones to develop azole-resistance, and the better ability of AR*Af* isolates to accept the azole-resistant genes via mating and recombination [11,14]. AR*Af* with TR_34_/L98H and TR_46_/Y121F/T289A mutations may have a genetic component that is restricting the resistance genotype to certain strains [24]. It has also been reported that the genetic backgrounds of TR_34_/L98H and TR_46_/Y121F/T289A AR*Af* were less diversified than those of wild-type isolates [24]. This is compatible with selective gene sweeps accompanying the selection of beneficial mutations and the genetic adaptation of *A. fumigatus* which enable it to survive and reproduce in prevailing or new environments, such as those with azole fungicides [24,25]. Selective sweeps would reduce allelic diversity and one, or a limited number, of clones would predominate locally [23]. The lower allelic and STR*Af* genotype diversity reported for AR*Af* isolates from sawmills in our study make this hypothesis likely. So, according to their genetic background, some strains could adapt more easily than others to a given environment and could be more able to develop and persist in an environment where the azole fungicide selection pressure is substantial. Moreover, the lower diversity of the AR*Af*, despite their different origin and the fact that they have been isolated in different sawmills, could suggest that AR*Af* belonging to the same group (with the Bruvo’s distance cutoff used) are genotypically closed and probably evolved from each other.

In conclusion, in this study we reported both common and different genotypes on a regional scale. Despite having lower allelic diversity than AS*Af* isolates, some AR*Af* could not be grouped together with other sawmill AR*Af* isolates. The diversity of genotypes described here, at a local level, seems to support the multiple origins hypothesis, thus suggesting that the hypothesis of clonal expansion from a common strain is now probably obsolete and insufficient to explain AR*Af* emergence and distribution. The evolution of AR*Af* from different origins could also be involved.

## Figures and Tables

**Figure 1 jof-06-00120-f001:**
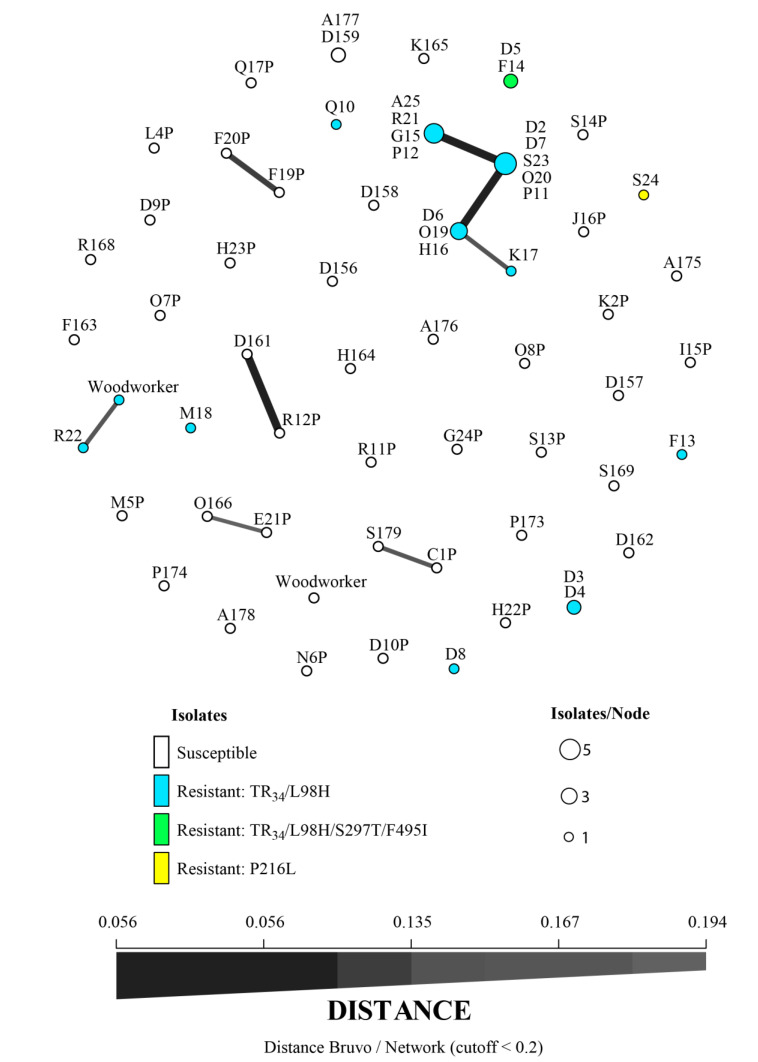
Minimum spanning network of nine microsatellite loci STR*Af* using Bruvo’s distance. Letters correspond to sawmill and the following numbers ± letters are genotype numbers. Each circle represents one multilocus genotype, the size of which is proportional to frequency. Different colors represent the different isolates (susceptible or resistant, and their mutations on the *Cyp51A* gene and its promoter*)*. Link thickness is proportional to genotype similarity.

**Table 1 jof-06-00120-t001:** Characteristics of analyzed azole-resistant (AR*Af*) and azole-susceptible (AS*Af*) *A. fumigatus*.

Isolate Identification	Date of Isolation	Origin	Susceptible or Resistant and Cyp51A Mutation
A25	February 2015	Substrate, sawmill A	Resistant, TR_34_/L98H
D2	January 2016	Substrate, sawmill D	Resistant, TR_34_/L98H
D3	January 2016	Substrate, sawmill D	Resistant, TR_34_/L98H
D4	January 2016	Substrate, sawmill D	Resistant, TR_34_/L98H
D5	January 2016	Substrate, sawmill D	Resistant, TR_34_/L98H/S297T/F495I
D6	January 2016	Substrate, sawmill D	Resistant, TR_34_/L98H
D7	January 2016	Substrate, sawmill D	Resistant, TR_34_/L98H
D8	January 2016	Substrate, sawmill D	Resistant, TR_34_/L98H
F13	February 2016	Substrate, sawmill F	Resistant, TR_34_/L98H
F14	February 2016	Substrate, sawmill F	Resistant, TR_34_/L98H/S297T/F495I
G15	February 2016	Substrate, sawmill G	Resistant, TR_34_/L98H
H16	February 2016	Substrate, sawmill H	Resistant, TR_34_/L98H
K17	March 2016	Substrate, sawmill K	Resistant, TR_34_/L98H
M18	March 2016	Substrate, sawmill M	Resistant, TR_34_/L98H
O19	March 2016	Substrate, sawmill O	Resistant, TR_34_/L98H
O20	March 2016	Substrate, sawmill O	Resistant, TR_34_/L98H
P11	September 2014	Substrate, sawmill P	Resistant, TR_34_/L98H
P12	September 2014	Substrate, sawmill P	Resistant, TR_34_/L98H
Q10	January 2016	Substrate, sawmill R	Resistant, TR_34_/L98H
R21	April 2016	Substrate, sawmill S	Resistant, TR_34_/L98H
R22	April 2016	Substrate, sawmill S	Resistant, TR_34_/L98H
S23	April 2016	Substrate, sawmill T	Resistant, TR_34_/L98H
S24	April 2016	Substrate, sawmill T	Resistant, TR_34_/L98H
Wood-worker	October 2013	Clinical	Resistant, TR_34_/L98H
A175	February 2015	Substrate, sawmill A	Susceptible
A176	February 2015	Substrate, sawmill A	Susceptible
A177	February 2015	Substrate, sawmill A	Susceptible
A178	February 2015	Substrate, sawmill A	Susceptible
C1P	January 2016	Substrate, sawmill C	Susceptible
D156	January 2016	Substrate, sawmill D	Susceptible
D157	January 2016	Substrate, sawmill D	Susceptible
D158	January 2016	Substrate, sawmill D	Susceptible
D159	January 2016	Substrate, sawmill D	Susceptible
D161	January 2016	Substrate, sawmill D	Susceptible
D162	January 2016	Substrate, sawmill D	Susceptible
D9P	January 2016	Substrate, sawmill D	Susceptible
D10P	January 2016	Substrate, sawmill D	Susceptible
E21P	January 2016	Substrate, sawmill E	Susceptible
F19P	February 2016	Substrate, sawmill F	Susceptible
F20P	February 2016	Substrate, sawmill F	Susceptible
F163	February 2016	Substrate, sawmill F	Susceptible
G24P	February 2016	Substrate, sawmill G	Susceptible
H22P	February 2016	Substrate, sawmill H	Susceptible
H23P	February 2016	Substrate, sawmill H	Susceptible
H164	February 2016	Substrate, sawmill H	Susceptible
I15P	February 2016	Substrate, sawmill I	Susceptible
J16P	March 2016	Substrate, sawmill J	Susceptible
K2P	March 2016	Substrate, sawmill K	Susceptible
K165	March 2016	Substrate, sawmill K	Susceptible
L4P	March 2016	Substrate, sawmill L	Susceptible
M5P	March 2016	Substrate, sawmill M	Susceptible
N6P	March 2016	Substrate, sawmill N	Susceptible
O7P	March 2016	Substrate, sawmill O	Susceptible
O8P	March 2016	Substrate, sawmill O	Susceptible
O166	March 2016	Substrate, sawmill O	Susceptible
P173	September 2014	Substrate, sawmill P	Susceptible
P174	September 2014	Substrate, sawmill P	Susceptible
Q17P	January 2016	Substrate, sawmill R	Susceptible
R11P	April 2016	Substrate, sawmill S	Susceptible
R12P	April 2016	Substrate, sawmill S	Susceptible
R168	April 2016	Substrate, sawmill S	Susceptible
S13P	April 2016	Substrate, sawmill T	Susceptible
S14P	April 2016	Substrate, sawmill T	Susceptible
S169	April 2016	Substrate, sawmill T	Susceptible
S179	April 2016	Substrate, sawmill T	Susceptible
Wood-worker	July 2013	Clinical	Susceptible

**Table 2 jof-06-00120-t002:** Polymorphism information for the nine microsatellite markers obtained after short tandem repeat for *A. fumigatus* (STR*Af*) typing of sawmill azole-resistant (AR*Af*) and azole-susceptible (AS*Af*) *A. fumigatus.*

Azole-Resistant *A. fumigatus* (AR*Af*, *n* = 24)					Azole-Susceptible *A. fumigatus* (AS*Af*, *n* = 42)				
Microsatellite Marker	N Alleles	Repeat Range	Median Size	Diversity (*D*)	Microsatellite Marker	N Alleles	Repeat Range	Median Size	Diversity (*D*)
2A	5	13–26	14	0.507	2A	11	10–27	18	0.768
2B	4	10–24	21	0.462	2B	9	12–25	19	0.766
2C	4	8–16	8	0.552	2C	11	8–20	12	0.868
3A	11	10–119	32	0.872	3A	19	10–49	26	0.909
3B	3	8–11	8	0.403	3B	10	8–22	9	0.771
3C	7	6–32	6	0.649	3C	19	6–45	18	0.924
4A	5	5–18	8	0.361	4A	13	7–26	9	0.774
4B	3	7–10	10	0.392	4B	8	5–26	9	0.746
4C	5	5–30	20	0.465	4C	9	5–36	7	0.770
Total alleles	47	NR	NR	NR	Total alleles	109	NR	NR	NR
Average D	NR	NR	NR	0.518	Average D	NR	NR	NR	0.811

N Alleles = Allele counts; *D* = Simpson index of diversity; NR = non relevant.

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
