# Peer review of "Molecular Epidemiology of Azole-Resistant Aspergillus fumigatus in Sawmills of Eastern France by Microsatellite Genotyping"

_jof, 2020, doi:10.3390/jof6030120_

Round 1
Reviewer 1 Report
The authors aim to study the genetic profile of azole-resistant and azole-susceptible Aspergillus fumigatus strains isolated in a previous study by applying the short tandem repeat for A. fumigatus assay, in order to identify a potential clustering of the isolates.
The paper is clear and well described, the tecniques used have been previously tested and verified to be valid and consistent. The content is mostly a confirmation of previous findings or hypothesis from the same group or from different research groups.
Despite the novelty of the results is not stricking, this represent another piece of the puzzle in order to better understand the mechanisms underlying the insurgence of azole resistance in A. fumigatus strains.
Author Response
We would like to thank you for your reviewing and comments.
Reviewer 2 Report
This studies provides additional information on the relationship between ARaf isolates from different or the source of isolation (sawmills) using STRaf.
In materials and methods section 2.1 the origin of the isolates is described. The authors should describe there total set in a table in which also the codes of the windmills are described. Those are used later in fig 1 but were not introduced as such
In fig 1 R22 is linked to the patient, it is however not clear if this patient was working at the same windmill (R). This could link the isolate in the patient with the presence of R22 in that environment.
The authors conclude that clonal expansion froma common strain seems not the only hypothesis, as indicated multiple origins hypothesis seems also involved. The author could in addition also conclude that evolution of from different origins is also involved, this is evidenced by the results in Fig 1 in which multiple isolates from different sources are linked as determined by their Bruvo genetic distance. that set of isolates are somehow related and evolved probably from each other. this could be discussed
Author Response
1) This study provides additional information on the relationship between ARaf isolates from different or the source of isolation (sawmills) using STRaf.
In materials and methods section 2.1 the origin of the isolates is described. The authors should describe there total set in a table in which also the codes of the windmills are described. Those are used later in fig 1 but were not introduced as such.
Authors’ response:
Thank you for your positive comments and your suggestions to improve our manuscript. Comments were taken into account. A table describing the characteristics and origins of analyzed azole-resistant A. fumigatus (ARAf) and azole-susceptible (ASAf) were added. We mentioned this table in the materials and methods section and called this “Table 1”. The Table 1, in the first version of the manuscript, corresponds to Table 2 in the new version.
2) In fig 1 R22 is linked to the patient, it is however not clear if this patient was working at the same windmill (R). This could link the isolate in the patient with the presence of R22 in that environment.
Authors’ response:
Isolate R22 was genotypically closed to the patient’s azole-resistant isolate but the patient did not work in this sawmill. Unfortunately, for legal reasons (work-related accident) we never knew which sawmill the patient worked in and we were unable to take samples from this sawmill. To clarify this point, we added the sentence “However the woodworker did not work in the sawmill R” in the result section.
3) The authors conclude that clonal expansion from a common strain seems not the only hypothesis, as indicated multiple origins hypothesis seems also involved. The author could in addition also conclude that evolution of from different origins is also involved, this is evidenced by the results in Fig 1 in which multiple isolates from different sources are linked as determined by their Bruvo genetic distance. That set of isolates are somehow related and evolved probably from each other. This could be discussed.
Authors’ response:
Comments were taken into account. We added, at the end of discussion section, the sentence “Moreover, the lower diversity of the ARAf despite of their different origin and the fact that they have been isolated in different sawmills, could suggest that ARAf belonging to the same group (with the Bruvo’s distance cutoff used) are genotypically closed and evolved probably from each other.”.
Moreover, we completed the conclusion be the sentence “The evolution of from different origins could be also involved.”.
Reviewer 3 Report
The manuscript investigates the genotypes of both azole-resistant and azole susceptible Aspergillus fumigatus strains isolated from the specific environment of sawmills, including clinical strains obtained from woodworkers. The main results of this study revealed that different azole-resistant might have emerged independently, instead of only one that spreads by clonal expansion. These results enlighten the consequences of the massive use of antifungal compounds and their release in the environment by humans during agricultural activities. Although the study has been carried out in a particular niche, its results suggest that similar adaptations are occurring in other environments.
I only suggest correcting a few typos for final polishing:
Bruvo’s distance, in the sentence: Thirty-two of the 42 ASAf (76%) had their own MLG and could not be grouped with the Bruvo distance cutoff used.
Sawmills, in the sentence: Identical MLGs were thus found in different sawmills, but different MLGs were also found in one sawmill: five MLGs were identified among the seven sawmill D ARAf.
In fact and Bruvo’s, in the sentence: Infact, contrary to ASAf, more than half of ARAf (54%) belonged to the same group with the Bruvo distance cutoff used and seemed to be genotypically close.
i.e., in the sentence: It was recently reported that certain practices like composting (i.e. stockpiling plant waste) might be the key to resistance selection in A. fumigatus [8].
Sawmills, in the sentence: Curiously, these two ARAf shared the same MLG as other sawmill ARAf.
An MLG, in the sentence: Conversely, a MLG identified twice in a single sawmill (D) is not linked to any profile in the London database.
Number of, in the sentence: Selective sweeps would reduce allelic diversity and one or a limited number clones would predominate locally [23].
Author Response
All comments were taken into account and the corrections of typos have been made in the manuscript.